# Insights into Bone Morphogenetic Protein—(BMP-) Signaling in Ocular Lens Biology and Pathology

**DOI:** 10.3390/cells10102604

**Published:** 2021-09-30

**Authors:** Daisy Y. Shu, Frank J. Lovicu

**Affiliations:** 1Department of Ophthalmology, Schepens Eye Research Institute of Mass Eye and Ear, Harvard Medical School, Boston, MA 02114, USA; daisy_shu@meei.harvard.edu; 2School of Medical Sciences, The University of Sydney, Sydney, NSW 2006, Australia; 3Save Sight Institute, The University of Sydney, Sydney, NSW 2000, Australia

**Keywords:** bone morphogenetic protein (BMP), transforming growth factor-beta (TGFβ), ocular lens, cellular signaling, cataract, epithelial-mesenchymal transition

## Abstract

Bone morphogenetic proteins (BMPs) are a diverse class of growth factors that belong to the transforming growth factor-beta (TGFβ) superfamily. Although originally discovered to possess osteogenic properties, BMPs have since been identified as critical regulators of many biological processes, including cell-fate determination, cell proliferation, differentiation and morphogenesis, throughout the body. In the ocular lens, BMPs are important in orchestrating fundamental developmental processes such as induction of lens morphogenesis, and specialized differentiation of its fiber cells. Moreover, BMPs have been reported to facilitate regeneration of the lens, as well as abrogate pathological processes such as TGFβ-induced epithelial-mesenchymal transition (EMT) and apoptosis. In this review, we summarize recent insights in this topic and discuss the complexities of BMP-signaling including the role of individual BMP ligands, receptors, extracellular antagonists and cross-talk between canonical and non-canonical BMP-signaling cascades in the lens. By understanding the molecular mechanisms underlying BMP activity, we can advance their potential therapeutic role in cataract prevention and lens regeneration.

## 1. Introduction

The activity of bone morphogenetic proteins (BMPs) was described by orthopedic surgeon, Marshall Urist in 1965 when he found that they could induce ectopic bone formation in rodents [1]; however, it was not until the late 1980s that the responsible BMP proteins were identified and characterized [2,3,4,5]. Since their initial discovery, BMPs have been shown to exert pleiotropic effects on many tissues and processes beyond bone and osteogenesis, now recognized as multifunctional proteins belonging to the transforming growth factor-beta (TGFβ) superfamily [6,7,8,9]. To date, over twenty BMPs have been identified to play important roles in embryogenesis, organogenesis and maintenance of adult tissue homeostasis [10]. BMPs are involved in many vital physiological processes including cell proliferation, differentiation, inhibition of growth and maturation in different cell types, dependent on their cellular microenvironment. Given our current knowledge, it is not surprising that they have been more aptly referred to as “body morphogenetic proteins” [11].

In an ocular context, BMPs are essential for early eye specification and patterning of the retina and lens [12]. In this review, we focus specifically on the role of BMPs in the lens in both normal and pathological contexts. Firstly, we briefly introduce BMPs including their receptors, signaling cascades and antagonists. We then discuss the importance of BMPs during the phases of lens development from the initial induction of the lens ectoderm in embryogenesis to later lens fiber differentiation processes. We follow this with a discussion of the role of BMPs in promoting lens regeneration and in abrogating lens pathology, including its potential as a therapeutic for cataract prevention. We conclude by highlighting opportunities to fill the gaps in our current understanding of BMP-signaling in the lens and propose directions for future research.

## 2. Bone Morphogenetics Proteins (BMPs)

### 2.1. Synthesis of BMPs

BMPs are synthesized as large precursor molecules of approximately 400–525 amino acids in length, to form 30–38 kDa homodimer proteins, with an amino (N)-terminal secretory signal peptide, a pro-domain for folding, and a carboxyl (C)-terminal mature peptide with seven cysteine residues [13]. These residues at the protein core form the highly conserved TGFβ-like cysteine knot configuration [13]. The seventh cysteine is critical for its biological activity, enabling dimerization with a second monomer through a covalent disulfide bond [14].

BMP precursor molecules undergo many post-translational modifications before the mature form is secreted. Following cleavage of the signal peptide, the precursor protein is glycosylated and dimerizes [15]. Cleavage of the pro-domain by pro-protein convertases in the trans-Golgi network, generates N- and C-terminal fragments that are secreted into the extracellular space [16]. The C-terminal segment containing the mature dimeric BMP protein with the cysteine knot is capable of binding to its receptor [16], while the pro-domain plays a more regulatory role [10].

The mature dimeric BMP proteins can either be homodimers, comprising two similar disulfide-linked BMPs (e.g., BMP-4/BMP-4) or heterodimers comprising of two different BMPs (BMP-2/BMP-4) [17]. This flexible oligomerization pattern broadens the scope of BMP interactions with its receptors, leading to activation of numerous signaling pathways for different cellular functions [17].

### 2.2. Classification of BMPs

Based on amino acid sequences and functional differences, the BMP subfamily is divided into different subgroups: BMP-2/4, BMP-5/6/7/8, BMP-14/13/12 (GDF5/6/7), GDF8/11, BMP-9 (GDF2)/BMP-10, GDF1/3 and GDF10/BMP-3 [16,18,19,20]. It should be noted that BMP-1 does not belong to the TGFβ superfamily as it shares homology with a pro-collagen, C-proteinase [21]. Although their monikers imply that all BMP members are inducers of bone, some can act as inhibitors of bone formation [10]. For instance, BMP-3 is a negative regulator of bone density [22], and BMP-13 strongly inhibits bone formation [23].

From gene inactivation studies in mice, it is clear that BMPs are critical for the development of various organ systems beyond bone [18]. BMP-2 knockout mice die due to amnion/chorion defects, and highlight the importance of BMP-2 for cardiac development [24]. BMP-4 deficient mice show early defects in limb patterning [25], as well as thymus and parathyroid morphogenesis [26]. BMP-7 knockout mice also display defects in skeletogenesis [27], as well as defects in neurogenesis [28], kidney [27], eye [27] and cardiac development [29]. In the adult, BMP-7 expression remains highest in the kidney [30,31,32], and to a lesser extent in cartilage [33], brain [34] and the eye [17]. Loss of BMP-3, BMP-5, BMP-6, BMP-8, GDF5/6/7, GDF8, GDF10, or GDF11 does not cause lethality, emphasizing the functional redundancy of BMPs in skeletal, cardiac and limb development [18].

Although some BMP subgroups share overlapping functions, some individual members display unique functions [18]. For instance, in the BMP-5/6/7 subgroup, BMP-5 and BMP-7 share similar functions, with BMP-6 uniquely involved in iron hemostasis, stimulating expression of hepcidin, a key regulator of iron absorption [35,36].

### 2.3. BMP Receptors: Specificity and Activation

Members of the TGFβ superfamily bind to two types of serine/threonine kinase receptors (type I and type II receptors) [37]. Both type I and type II receptors share similar structural properties, comprised of a short extracellular domain of 10–12 cysteine residues, a transmembrane domain, and a cytosolic serine/threonine kinase domain [14]. The intracellular domains of type I receptors, but not type II receptors, have a characteristic glycine and serine-rich domain (GS domain) located N-terminally to the serine/threonine kinase domains [37]. Both types of receptors are required to form a functional complex to propagate downstream signaling events [17,38,39].

While TGFβ binds exclusively to its type I receptor, TGFBR1 (activin receptor-like kinase (ALK)-5 or TβRI) and type II receptor, TGFBR2, BMPs have five type I receptors; Acvrl1 (also known as ALK1), ActRI (ALK2), BMPR-IA (ALK3), ActRIb (ALK4) [40] and BMPR-IB (ALK6), and three type II receptors; BMPR-II, ActRIIa, and ActRIIb [14]. BMPR-II is specific for BMPs, whereas ActRIIa and ActRIIb are also shared by activins and myostatin [37]. Differing affinities for the various BMP molecules and their preferred ligand-receptor complexes have been identified (summarized in Figure 1) [37,41].

In general, ligand binding of TGFβ superfamily members induces the constitutively active serine/threonine domains of type II receptors to transphosphorylate the GS domain of the type I receptor, forming a heterotetrameric complex [37]. In contrast, the binding of BMP-2 in particular, follows a different sequential binding mechanism [42,43], with BMP-2 first binding to its type I BMP receptor (high affinity receptor) that then activates recruitment of the type II BMP receptor (low affinity receptor) into a ternary complex [42], similar to TGFβ. Type I and type II BMP receptors can independently bind BMP-2, but in the presence of both receptor types, there is enhanced binding affinity [43,44,45].

### 2.4. BMP Intracellular Signaling Pathways

BMPs can activate different signaling pathways through distinct receptor complexes (summarized in Figure 2) [46]. For example, BMP-2 has been shown to have two modes of signal transfer; (i) BMP-2 binds to a preformed complex (PFC) of BRIa and BRII that triggers clathrin-mediated endocytosis and initiates the canonical Smad-signaling pathway [43,47]. (ii) BMP-2 binds to its high affinity receptor BMPR-IA, upon which BMPR-II is recruited into the complex, forming a BMP-induced signaling complex (BISC) [48] resulting in its internalization via caveolae and activation of the non-Smad, mitogen-activated protein kinase (MAPK) pathway [49].

#### 2.4.1. Canonical Signaling Pathway

The canonical BMP-signaling pathway involves the small mothers against decapentaplegic (Smad) proteins [50]. Smads are proteins that mediate intracellular signals and regulate gene transcription of TGFβ and BMP target genes. Based on their function, they are divided into three classes of Smads: the receptor-regulated Smads (R-Smads), the common-mediator Smads (Co-Smads) and the inhibitory Smads (I-Smads) [37]. The activated receptor complex relays the signal to the cytoplasm by phosphorylating the carboxy-terminus of receptor-regulated Smad proteins (R-Smads) [51]. R-Smads of the TGFβ/activin pathway include Smad2 and Smad3, whereas Smad1, Smad5 and Smad8 participate in BMP-signaling [37]. Similar to the Smad anchor for receptor activation (SARA) cofactor in TGFβ-signaling that interacts directly with and recruits Smad2/3 to the TGFβ receptor [52], the Smad1 anchor for receptor activation for BMP-signaling is endofin, that enhances Smad1 phosphorylation and its translocation to the nucleus [53].

Phosphorylated R-Smads hetero-oligomerize with Smad4, a Co-Smad shared by both TGFβ- and BMP-signaling [18]. This complex translocates to the nucleus, binding to the Smad-binding element (SBE), or BMP-responsive element (BRE), to regulate transcription of respective target genes [50]. As Smads have a lower intrinsic binding affinity to DNA, they cooperate with transcriptional co-activators or co-repressors, and chromatin remodeling factors, to facilitate the integration of different signaling inputs, accounting for the multitude of gene responses generated by the few Smad proteins [18].

The inhibitory I-Smads (Smad6 and Smad7) can interrupt phosphorylation of R-Smads by negatively regulating Smad activation [54]. Their absent SSXS phosphorylation site allows Smad6 and Smad7 to form stable associations with the activated type I receptors, preventing subsequent phosphorylation of R-Smads and Co-Smads [10]. Smad7 can inhibit both TGFβ- and BMP-signaling, while Smad6 inhibition is specific to BMP-signaling [55]. Smad6 can also inhibit signaling by acting as a transcriptional co-repressor and competing with R-Smads for Co-Smad binding [49]. Furthermore, I-Smads have been found to mediate receptor interaction with E3-ubiquitin ligases; Smad6 and Smad7 facilitate Smad ubiquitin regulatory factors (Smurf)1 and Smurf2 ubiquitinating and degrading R-Smads and BMP receptors [56]. Smad6 and Smad7 expression can be upregulated by TGFβ, activin and BMP, suggesting that I-Smads function in a negative feedback loop to antagonize both TGFβ- and BMP-signaling [49]. Moreover, TGFβ, activin and nodal pathways can also interact with BMP type I receptor to phosphorylate Smad2/3, hence diverting the canonical BMP-signaling pathway [57].

#### 2.4.2. Non-Canonical Signaling Pathway

In addition to the canonical signaling cascade, BMP can also signal through several non-canonical, Smad-independent pathways [49]. These include the MAPKs, p38 and the extracellular signal-regulated kinase (ERK), C-Jun N-terminal kinase (JNK), nuclear factor-kappa beta (NF-κB) [14] and PI3K/Akt pathways [58,59,60]. Activation of the non-Smad pathways is believed to be through the interactions with BRAM1 (bone morphogenetic protein-receptor-associated molecule 1) and XIAP (X-linked inhibitor of apoptosis protein), and downstream molecules such as TAK1 (TGFβ-activated kinase 1) and TAB1 (TAK1 binding protein), that form the TAB1-TAK1 complex [14]. Integration and cross-talk of diverse non-Smad and Smad pathways broadens the cellular responses elicited by BMP, and is a key mechanism for modulation of specific developmental responses [61,62].

### 2.5. Antagonists of BMP-Signaling

The specificity, intensity, and duration of BMP-signaling is regulated on multiple levels by extracellular and intracellular modulators ranging from interaction of the ligand with secreted antagonists, crosstalk with other signaling cascades, or modes of receptor oligomerization and internalization [10]. Several secreted extracellular antagonists modulate the activity of BMP at the cell surface by preventing its binding to its receptor complex (reviewed by Massague and Chen) [61,63]. BMP antagonists also possess a cysteine knot structure and according to the size of their cysteine knot, they have been classified into three subfamilies: the CAN family (eight-membered ring); twisted gastrulation protein (nine-membered ring); and chordin and noggin (ten-membered ring) [64]. The CAN family is further subdivided into Gremlin/DRM/IHG-2, Cerberus, Coco, DAN, protein related to DAN and Ceberus (PRDC), Sclerostin and USAG-1 [64].

BMP antagonists exhibit different binding affinities for various members of the BMP family [65]. Noggin binds BMP-2 and BMP-4 with 10–15 times greater affinity than the BMP receptors, effectively abolishing the activity of BMP-2 and BMP-4 [66]. Noggin also binds to BMP-7, but with lower affinity [63]. Interestingly, BMP is capable of inducing noggin expression and initiating a negative feedback loop to limit its own activity [67,68,69]; however, BMP-6 and BMP-9 are naturally insensitive toward noggin [70,71]. Chordin binds BMP-2 and BMP-4 with 10 times lower affinity compared to noggin [66]. The chordin/BMP complex may be cleaved by BMP-1 metalloproteinase to release biologically active BMP, suggesting a complex regulation of BMP interaction with its receptors [72].

Intracellular antagonists of BMP-signaling include I-Smads, microRNAs, such as miR-21 that negatively regulates BMP-4 [73], phosphatases, such as PP1 and PP2A that dephosphorylate the BMP receptors, and R-Smad and FK506-binding protein 1A that binds to the GS domain of type I receptors to inhibit receptor internalization [63]. Co-receptors in the plasma membrane, such as endoglin, betaglycan and the repulsive guidance molecule (RGM) family including RGMa, RGMb (also known as Dragon), RGMc (also known as hemojuvelin or HJV), and RGMd, modulate the interactions between BMP ligands and type I and II BMP receptors to enhance the level of regulation [74,75]. BMP-signaling can also be blocked by the pseudo-receptor BAMBI (BMP and activin membrane bound inhibitor), a transmembrane protein with an extracellular domain similar to that of type I BMP receptors [76]. BAMBI’s inhibitory effects are mediated by a short intracellular domain that lacks the serine/threonine-kinase segment, thus preventing the formation of receptor complexes and subsequent BMP-signaling [76].

## 3. Role of BMP-Signaling in Lens Development

The vertebrate lens is an ideal model system for studying organ morphogenesis and cell differentiation due to the ease of manipulation and visualization of lens tissues [77]. Since Hans Spemann introduced the concept of inductive interactions during the study of lens development in 1901, developmental biologists have used the lens as a tool to elucidate the general molecular mechanisms underlying embryonic induction, cell specification, and patterning of different tissues and organs [78].

The development of the eye involves a hierarchy of inductive interactions between the embryonic forebrain and the overlying surface head ectoderm (see McAvoy 1980) [79]. Briefly, lens development is morphologically first seen as a thickening of the embryonic surface head ectoderm into the lens placode, apposed to the optic vesicle [78]. The lens placode invaginates into the optic vesicle to form the lens pit that then deepens to pinch off from the surface ectoderm (the prospective cornea), to form a hollow structure, the lens vesicle [80]. The posterior lens vesicle cells elongate and differentiate into primary lens fibers filling the vesicle lumen [80]. The anterior lens vesicle cells go on to form a monolayer of lens epithelial cells that continuously proliferate and subsequently differentiate at the lens equatorial region into secondary fiber cells [80]. The differentiation of lens epithelial cells into secondary lens fiber cells is characterized by extensive cell elongation and accumulation of specialized crystallins [81]. All intracellular membrane-bound organelles are eventually degraded, with cessation of DNA, RNA and protein synthesis [81]. The process of lens fiber differentiation continues throughout life, forming a mass of secondary fiber cells arranged in concentric layers surrounding a dense central nucleus of mature primary fibers [82]. A summary of the studies investigating the role of BMPs in different aspects of lens development (Figure 3), regeneration and pathology is provided in Table 1.

### 3.1. Lens Specification

The optic vesicle plays a key role in lens formation by providing inductive signals to the surface ectoderm to form the lens placode [78]. BMPs have been identified as putative signaling molecules contributing to this inductive event [80,111]. At the gastrula stage, BMP-2 and BMP-4 have been detected at the anterior neural plate border [112,113] where both prospective lens and olfactory progenitor cells are located [114]. BMP downstream mediators, phospho-Smad1/5/8, have also been identified in this region, indicating active BMP-signaling [112,113]. In mice, expression of BMP-4 rapidly decreases in the presumptive lens ectoderm by embryonic day 9.5 (E9.5), and is completely absent in the lens placode at E10 [83,84]. Similarly, BMP-7 is also expressed in the head surface ectoderm at E9.5, but slowly diminishes in the lens placode by E10, before becoming absent from the lens vesicle by E10.5 [115].

At the gastrula stage, prospective lens and olfactory placodal cells intermix in a domain at the rostral neural plate border [114]. The spatial separation of lens and olfactory progenitor cells occurs at the neural fold stage and by early neural tube stages, the presumptive lens ectoderm overlies the optic vesicle [116]. Sjödal et al. (2007) showed that BMP activity is both required and sufficient to induce lens and olfactory placodal cells. Prospective forebrain explants from chick embryos in the gastrula stage, cultured in the presence of BMP-4, generated cells of an olfactory and lens placodal character [86]. Continued exposure of placodal progenitor cells to BMP signals resulted in lens specification whilst olfactory placodal cells were generated once BMP signals were downregulated. Hence, temporal changes in BMP activity can act as a switch in establishing olfactory and lens placodal identity. The concentration of BMP activity also plays a crucial role. Exposure of prospective rostral border cells to a higher level of BMP-signaling (>50 ng/mL) promoted an epidermal cell identity and repressed neural cell fate [86]. Conversely, culturing these prospective lens and olfactory cell explants in the presence of noggin generated cells of neural forebrain character [86]. This is consistent with the theory that BMP-activity suppresses neural fate and varying the temporal onset and concentration of BMP-signaling can modulate the differential specification of olfactory, lens and epidermal cell fates.

Pandit et al. (2011) further explored the temporal requirement of BMP during early lens development in relation to L-Maf, a lens-specific member of the Maf family of transcription factors. During the lens placodal stage, L-Maf expression is upregulated in chick [80], and C-Maf in mouse [117]. Following this, an early step of primary lens fiber differentiation involves the upregulation of crystallin proteins, including δ-crystallin in chick [77]. In the developing lens ectoderm, BMP-4 and pSmad1/5/8 expression precedes the onset of both L-Maf and δ-crystallin expression [96]. While BMP activity is both required and sufficient to induce L-Maf expression, the subsequent cell elongation and upregulation of δ-crystallin occurs independently of further BMP-signaling. These results extend the knowledge of lens development and cell fate, highlighting the role of BMP in lens specification and subsequent BMP-induced L-Maf as a regulator of early differentiation of primary lens fiber cells.

Huang et al. (2015) showed that autoregulation of BMP-signaling is a key molecular mechanism underlying lens specification [89]. BMP inhibition by targeted deletion of type I BMP receptors, Bmpr1a and Acvr1, in murine lens-forming ectoderm, and exposure of chick pre-lens ectodermal explants to noggin, resulted in an upregulation of *Bmp2* and *Bmp4* transcripts to generate olfactory cells [89]. Conversely, exposure to BMP-4 lowered expression of *Bmp2* and *Bmp4* transcripts resulting in characteristic epidermal cells [89]. This agrees with previous studies showing that lens specification requires continued BMP activity and that high levels of BMP signals promote epidermal specification [86,96]. Hence, an intermediate and balanced level of BMP activity is required for lens specification, and a reduction or increase in BMP activity can result in the generation of olfactory placodal or epidermal cells, respectively [89]. Exposure of chick ectoderm explants to noggin did not affect *Bmp7* levels; however, addition of BMP-7 increased expression of *Bmp7* transcripts, indicating positive autoregulation of BMP-7-signaling in the chick pre-lens ectoderm [89]. In contrast, blocking BMP-signaling (by deletion of type I BMP receptors) in mice resulted in an increase in *Bmp7* expression [89]. This discrepancy in Bmp7 autoregulation may be attributed to differences in animal models, but also to temporal differences in upregulation of *Bmp4* and *Bmp7* transcripts in mice as *Bmp4* expression increases much faster compared to *Bmp7* [89]. Hence, long-term inhibition of BMP-signaling may be necessary to upregulate *Bmp7* expression in chick. Future studies are required to understand this discrepancy and elucidate the signaling pathway(s) responsible for BMP autoregulation in the pre-lens ectoderm.

### 3.2. Lens Induction

Disruption to the BMP pathway consistently led to disturbances in lens induction [83,84,118]. In BMP-4 knockout mice, there was no lens induction despite the close contact between the head ectoderm and optic vesicle, with concurrent loss of *Sox2* expression, a transcriptional regulator of crystallin genes in early lens fiber differentiation [83,119]. Lens formation and *Sox2* expression could be restored in BMP-4 null mutant embryo tissues by exogenous application of BMP-4-soaked beads to the optic vesicle in explant cultures; however, replacement of the optic vesicle in wild-type mouse eyes with BMP-4-carrying beads, or other *Bmp4-*expressing tissues, was not able to induce lens formation or *Sox2* expression in head ectoderm, indicating that BMP-4 alone is not sufficient to mimic the inductive properties of the optic vesicle. These results suggest that BMP-4 may regulate induction by acting synergistically with additional factors expressed within the optic vesicle. Since BMP-4 is secreted, the use of tissue recombination techniques is limited in elucidating its function separately in the ectoderm and optic vesicle, and hence, future studies should address this by inhibiting BMP-4-signaling in a cell-type-specific manner during lens induction.

Another BMP family member, BMP-7, is also expressed in regions of the early developing eye that partially overlap with BMP-4 [115]. BMP-7 is present in both the optic vesicle and the surface ectoderm at the time of lens placode thickening and is crucial in the early lens induction process [84,120]. While *Bmp4* null mice consistently showed an absence of lens formation in all cases, variability in the phenotype of *Bmp7* null mice is evident, with mice displaying unilateral or bilateral eye defects [84,115,121]. The majority (60%) of BMP-7-deficient embryos displayed profound bilateral deterioration of the developing retina, optic nerve and lens, while the remaining 40% exhibited either unilateral or bilateral microphthalmia with morphologically normal ocular structures but half their normal size [115]. The variable penetrance of eye abnormalities may be attributed to the rapidly changing expression levels of BMP-7, between E9.5 and E11 [84]. In *Bmp7* null mice, the expression of *Pax6*, an essential transcription factor for early eye development, was maintained in the optic vesicle but no longer detected in the surface ectoderm [84]. These results indicate that BMP-7-signaling is required for the maintenance of *Pax6* expression in the prospective lens placode ectoderm, but not for its initial induction. It is likely that a linear pathway exists in that BMP-7 functions upstream of *Pax6* to regulate lens placode induction.

### 3.3. Lens Placode Invagination

The invagination of the lens placode to become the lens pit involves a series of molecular and cellular processes including cell proliferation, cell crowding and cytoskeletal reorganization [122]. Initially, cell proliferation in the thickening lens placode results in cell crowding [122], with a redistribution of components of the actin cytoskeleton including filamentous actin (F-actin) and tight junction proteins, such as zonular occludens (ZO)-1 [123]. With placode invagination, phalloidin staining for F-actin decreases along the lateral surfaces of cells and increases at their apical ends, with the apical distribution of ZO-1 remaining continuous [88]. This is consistent with the “drawstring” mechanism for tissue invagination that proposes that the contraction of apical F-actin filaments draws the apical ends of cells together to enable bending of the placode to form the lens pit [88]. The importance of BMP-signaling in regulating this “drawstring” mechanism for lens placode invagination is highlighted in mice lacking both BMP receptors, *Bmpr1a* and *Acvr1,* using a Pax6-Cre transgene, LeCre [88]. Here, the lens did not form, with F-actin remaining uniformly distributed at the cell periphery, not accumulating at the apical ends of the lens placode cells. Concurrently, ZO-1 remained discontinuous at the apical ends of the cells suggesting the absence of apical contraction. Interestingly, deletion of the genes encoding the canonical transducers of BMP-signaling, Smad1, Smad5 and Smad4 did not affect apical re-localization of F-actin and these mice were able to form lenses, suggesting that actin cytoskeleton reorganization is regulated by Smad-independent BMP-signaling. Upregulation of the expression of lens-specific markers, including FoxE3 (transcription factor), and αA-crystallin (an abundant structural lens protein), were also found to be regulated by BMP receptors in a Smad-independent manner [88]. Moreover, Yoshimoto et al. (2005) showed that FoxE3 is indirectly dependent on Smad-interacting proteins, specifically, Smad8 augments Smad interacting protein-1 (Sip1)-activity, a transcription factor upstream of FoxE3. Notably, based on mouse dataset mining (iSyte), Smad8 is one of the select Smad members not found in the developing lens (from E10.5) nor postnatal lens [124]. Further studies are required to define the alternative downstream BMP Smad-independent pathways mediating lens placode invagination, and the initial upregulation of lens-specific markers.

Both Bmpr1 and Acvr1 play redundant roles as either receptor is sufficient for lens formation [88]. Despite their redundancy, these two BMP type I receptors display unique functions in lens development. Bmpr1a promotes the survival of placode lens cells, while Acvr1 promotes cell proliferation. Such distinct functions of these receptors in the lens appears to be mediated by differing downstream signaling pathways. Promotion of cell survival involves R-Smads, Smad1 and Smad5, whereas cell proliferation is regulated by one or more Smad-independent pathways. Future studies should examine which BMP ligands are responsible for eliciting these distinct responses from the type I BMP receptors. Interestingly, Smad4 is not required for cell survival or proliferation in the lens placode [88], suggesting that R-Smads may bind to factors other than Smad4 to mediate BMP-signaling.

Targeted deletion of type I BMP receptors from the pre-lens ectoderm using LeCre not only prevented lens formation, but also resulted in coloboma-like defects, highlighting the importance of BMP activity for the closure of the optic cup [89]. Lens placode invagination occurs in concert with the invagination of the optic vesicle to form the lens pit and optic cup, respectively. The interplay between inductive signals from the presumptive retina and lens remains unclear, and further research in this area will shed light on the complexities of the ocular morphogenesis machinery.

### 3.4. Lens Fiber Differentiation

#### 3.4.1. Role of FGF in Lens Fiber Differentiation

Since the seminal work of the McAvoy laboratory in the 1980s, it is now widely accepted that members of the fibroblast growth factor (FGF) family play a central role in lens fiber differentiation [82,125]. In vitro studies provided compelling evidence that FGF was the only growth factor with the ability to induce mammalian lens epithelial cells to undergo fiber-specific morphologic [126,127] and molecular changes [125] including cell elongation, structural membrane specialization and initiation of specific crystallin gene expression. This was further supported by in vivo studies where overexpression of a dominant-negative FGF receptor in transgenic mice [128,129,130] and conditional deletion of FGF receptors (*Fgf1-3*) [131] both led to the inhibition of fiber differentiation, elegantly highlighting the importance of FGF receptor signaling in regulating lens fiber differentiation.

#### 3.4.2. Role of BMP Ligands in Lens Fiber Differentiation

Although there is convincing evidence that FGF signaling is required for lens fiber differentiation, FGFs alone cannot account for all the fiber differentiation-activity of the vitreous humor [82]. There has been growing evidence that other ocular growth factors, in particular, BMPs, are able to enhance the synthesis of fiber-specific proteins (reviewed in Lovicu and McAvoy, 2005) [82]. BMP-4 and BMP-7 on organ cultures of embryonic chick lens placodes and optic vesicles enhanced lens growth and expression of the fiber differentiation marker, δ-crystallin [132]. Boswell et al. (2008) also found that exogenous BMP-2, -4 and -7 upregulated both morphological features and biochemical markers of fiber differentiation, including δ-crystallin and CP49, in dissociated cell-derived monolayer (DCDML) cultures from primary embryonic chick lens epithelial cells [81]. In contrast, two previous studies in vitro that examined the effect of BMPs on chick [92] and rat [133] lens epithelial cells did not find any evidence to show that BMPs could enhance the morphological differentiation or the expression of fiber cell marker proteins. This may be due to differences in model systems as both these groups used central lens epithelial explants, whereas Boswell et al. (2008) utilized embryonic DCDML cultures that include peripheral epithelial (pre-equatorial and equatorial) cells that are more responsive to differentiation stimuli compared to central epithelial cells [127,134]. Since epithelial-to-fiber cell differentiation is localized to the peripheral regions of the lens in situ, models such as DCDML cultures and whole lens epithelial explants, are a more physiologically relevant model system for recapitulating the process of fiber differentiation [134].

Hung et al. (2002) overexpressed BMP-7 in lenses of transgenic mice that resulted in widespread apoptosis and ablation of the neural retina [90]. This process occurred rapidly such that only a small fraction of the neural retina remained by E15.5 and disappeared altogether by postnatal day 1 (P1). Interestingly, retinal ablation was correlated to shifting of the lens bow region posteriorly until the LECs completely surrounded the lens, highlighting the importance of the retina in providing positional lens fiber cell differentiation cues. In these mice, when FGF-3 was overexpressed in lens, this rescued the loss of fiber cell differentiation, indicating that BMP-7 overexpression in the lens does not incapacitate the ability of LECs to respond to differentiation signals. Consistent with these findings, Pandit et al. (2015) showed that BMP signals emanating from the lens are critical for the specification of neural retinal identity and induction of neural retinal cells [135]. Further studies are required to characterize the crosstalk between lens and retina in providing complementary survival and differentiation cues to each other. French et al. (2009) extended the spectrum of BMP molecules that affect lens fiber differentiation to include GDF6a. Knockdown of *gdf6a* in zebrafish resulted in the absence of pSmad1/5/8 in the lens and downregulation of multiple lens-specific genes including *cryba2a* and *lim2.3* [87]. The addition of dorsomorphin, a Bmp-signaling inhibitor, disrupted lens fiber cell differentiation. Hence, in the zebrafish eye, lens fiber development requires both GDF6a and other sources of BMP-signaling that are yet to be elucidated.

BMP-4 and its receptors have been detected in the adult rat eye, showing abundant and differential expression in various ocular structures including the cornea, iris, ciliary body, lens and retina [136]. Specifically, in the lens, BMP-4 and its receptors BMPR-IA, BMPR-IB and BMPR-II were identified in lens epithelial cells and lens cortical fiber cells; however, they were not expressed in the central region of the lens [136]. Therefore, in addition to regulating primary lens fiber differentiation, the abundance of BMP-4 and its receptors indicate a role for BMP-signaling in secondary lens fiber differentiation in adult life.

#### 3.4.3. Role of BMP Antagonists in Lens Fiber Differentiation

Consistent with the BMP culture studies, Faber et al. (2002) highlighted the importance of BMP-signaling in primary lens fiber differentiation using noggin, a BMP ligand antagonist [91]. The addition of noggin to organotypic cultures of E10.5 mouse whole eye explants resulted in smaller lenses, mostly due to the reduction in primary fiber cell mass [91]. Beebe et al. (2004) corroborated these findings by showing that noggin partially inhibited epithelial cell elongation in embryonic chick lens epithelial explants, with higher levels unable to further inhibit this elongation [118]. Follistatin, an activin-binding protein antagonist, had no effect on cell elongation. Adding noggin and follistatin together; however, completely inhibited cell elongation, indicating that both BMP and activin contribute to lens fiber differentiation [118].

Injection of noggin-expressing retrovirus into optic vesicles of E2 chick embryos resulted in delayed lens fiber differentiation [92]. At E4, noggin-infected lenses displayed fiber cells that had not elongated or had only elongated slightly, and by E6, these fiber cells were essentially normal, apart from slightly retarded cell elongation at the lens equator. This highlights the importance of BMP in the earlier stages of lens fiber cell differentiation. Similarly, overexpression of noggin in the lenses of transgenic mice resulted in defects of the equatorial epithelial cells. Instead of forming a lens bow at the equator, the epithelial monolayer extended beyond this to the posterior lens with cells retaining a similar morphology to anterior epithelial cells, with no evidence of apoptosis, multilayering, elongation or even aberrant mesenchymal transdifferentiation [81]. Mice overexpressing noggin did display visibly smaller lenses than wild-type mouse controls, with 32% less total protein per lens at 2 weeks of age, and a striking reduction in the synthesis of all three major mammalian crystallin families, α, β and γ [81]. Taken together, these results emphasize the key requirement for BMP-signaling in secondary lens fiber differentiation [81]. A confounding issue acknowledged in these experiments is that noggin overexpression can affect other ocular structures, including loss of the vitreous body. Since the vitreous humor is considered the major reservoir of FGF for lens differentiation, the absence of fiber differentiation could be due to the compromised vitreous body.

#### 3.4.4. Role of BMP Receptors in Lens Fiber Differentiation

BMP receptors, ALK3, ALK6 and BMP receptor II, have been identified in the lens epithelium [90,93,137]. Beebe et al. (2004) showed that targeted deletion of ALK3 in the lens resulted in a small lens phenotype, with a thin epithelial layer by E13.5 that remained smaller than normal throughout development, indicating a role for ALK3-signaling in maintaining cell viability and/or proliferation [118]. The fiber cells appeared disorganized, vacuolated and degenerated by postnatal day 9, and in some cases the anterior capsule was ruptured [118]. Moreover, lenses lacking in ALK3 were surrounded by abnormal mesenchymal cells, with a condensed pigmented mass surrounding the hyaloid vasculature and hypercellular vitreous body. Despite specific targeted deletion of ALK3 in the lens, these lens extrinsic ocular defects suggest that aberrant signals from the lens may be negatively impacting other parts of the eye. Alternatively, a compounding factor may be the use of the Le-Cre transgene that is known to impact ocular tissues other than lens [138].

Immunoreactivity for BMP type 2 receptor and nuclear phosphorylated BMP-responsive Smads are localized to the equatorial cells of the lens vesicle, indicating the active role of BMP-signaling in these primary differentiating cells [91]. This is supported by the inhibition of primary fiber cell elongation at E13.5, when a dominant-negative form of the type I BMP receptor, ALK6, was overexpressed in the lenses of transgenic mice [91]. Interestingly, the observed inhibition of primary fiber differentiation was asymmetrical, appearing only in the ventral half on the nasal side of the lens, suggesting that distinct differentiation stimuli may be active in different quadrants of the eye [91]. As the lens continues to develop, the equatorial epithelial cells proliferate, migrate posteriorly and differentiate into secondary lens fiber cells. Belecky-Adams et al. (2002) identified the accumulation of pSmad1 in the nuclei of epithelial cells immediately before and at the beginning of their elongation into secondary lens fiber cells. The expression of pSmad1 later subsided in fiber cell elongation and was barely evident in deeper cortical lens fiber cells [92]. Anterior to the lens equator, epithelial cells show no nuclear staining for pSmad1, with Beebe et al. (2004) showing strong immunoreactivity for pSmad1 in nuclei of cells at the lens equator that decreased soon after the cells elongated [118]. In contrast, activin-induced upregulation of pSmad2 was absent at the lens equator, and appeared during lens fiber elongation, remaining strong throughout the later stages of lens fiber differentiation and maturation, signifying distinct roles for both BMP and activin in lens differentiation [118].

The type I BMP receptor, Acvr1, plays an important role in regulating lens cell proliferation and cell cycle exit during early fiber cell differentiation [88]. Using the *Acvr1* conditional knockout mouse (*Acvr1*^CKO^) model, Acvr1-signaling was found to promote proliferation in early stages of lens development. At later stages, however, Acvr1 inhibits proliferation of LECs in the transitional zone to promote cell cycle exit; a process necessary for the proper regionalization of the lens epithelium and subsequent secondary lens fiber differentiation. Acvr1-promoted proliferation was Smad-independent, whereas its ability to stimulate cell cycle exit was through the canonical Smad1/5-signaling pathway. Loss of Acvr1 also led to an increase in apoptosis of lens epithelial and cortical fiber cells, and together with the reduction in proliferation, led to a small lens phenotype in these *Acvr1*^CKO^ mice.

The fiber cells of the Acvr1 conditional knockout mouse exhibited increased nuclear staining for the tumor suppressor protein, p53 (encoded by *Trp53*) [97]. In double conditional knockout (*Acvr1;Trp53^DCKO^*) mice, loss of p53 reduced Acvr1-dependent apoptosis in postnatal lenses, indicating that p53 may be important for eliminating aberrant fibers that escape cell cycle exit [97]. As these surviving cells were deficient in BMP-signaling, they were unable to respond to signals promoting cell cycle withdrawal and thus, their continued proliferation led to tumor-like masses at the posterior of the lens that exhibited morphological and molecular similarities to human posterior subcapsular cataract (PSC) [97]. With age, these masses grew to the form vascularized tumors [97]. *Trp53^DCKO^* lenses also resulted in PSC-like changes; however, the cells in these plaques did not proliferate, unlike those in *Acvr1;Trp53^DCKO^* lenses [97]. These observations support the role of Acvr1 as a tumor suppressor in the lens, as concurrent loss of Acvr1 allows the aberrant fiber cells to escape the normal growth-inhibitory signals transduced by Acvr1-signaling.

#### 3.4.5. Synergistic Roles of FGFs and BMPs in Lens Fiber Differentiation

A balance of FGF and BMP signals is required to regulate the early differentiation of primary lens fiber cells in embryonic chick lens [94]. Equarin, a soluble protein, is upregulated in the early-formed lens vesicle before the formation of the first primary lens fiber cells, and its expression is subsequently restricted to sites of fiber differentiation at the lens equator [139]. BMP activity was found to induce Equarin, in a FGF-dependent manner [94]. Although FGF activity is necessary for the induction of Equarin expression, alone it is not sufficient [94]. For FGF-induced lens cell proliferation, in the absence of BMP-activity, cell cycle length was prolonged, or cells were arrested in the cell cycle, suggesting that a counterbalance of BMP- and FGF-activity is required to regulate cell cycle exit. Taken together, these results indicate that while FGF activity can regulate lens epithelial cell proliferation, BMP-signaling is required to promote cell cycle exit and early differentiation of primary lens fiber cells. Future studies are needed to investigate the downstream signaling pathways involved in this complex interplay of FGF- and BMP-activity.

The synergistic role of FGF and BMPs has also been demonstrated in secondary lens fiber differentiation. Boswell et al. (2008) showed that inhibition of BMP-activity with noggin or anti-BMP antibodies, prevented FGF from upregulating fiber differentiation markers including δ-crystallin, CP49 and filensin in DCDMLs [81]. This was further explored by Boswell et al. (2015) where noggin prevented FGF from stimulating FRS2, its docking protein constitutively bound to FGF receptors, indicating that BMP-activity is required at the level of FGF receptor activation. Interestingly, FGF promoted the expression of both BMP-4 and BMP target genes in lens cells [99], highlighting a novel mode of reciprocal cooperation between FGF and BMP pathways, whereby BMP keeps lens cells in an optimally FGF-responsive state, with FGF potentiating endogenous BMP-signaling by promoting BMP-mediated gene expression. This agonistic relationship between BMP and FGF may explain why disruption of either FGF or BMP signaling in the lens results in deleterious effects on lens development.

### 3.5. Gap Junction-Mediated Intercellular Communication in Lens Cells

Gap junctions are highly specialized intercellular channels that facilitate the exchange of ions, low molecular mass (<1 kD) second messengers, and nutritional metabolites between functionally and structurally distinct regions of tissues, including the lens [140]. Due to its avascularity, a network of gap junctions is required in facilitating the lens syncytium, permitting both electrical and biochemical coupling between cells. The anterior lens epithelial cells are in closer contact with nutrients of the aqueous humor, providing the metabolic energy to maintain correct ion and metabolite concentrations within the lens fiber mass, hence maintaining tissue homeostasis and thus, lens transparency [140]. Mature fiber cells contain a significantly large number of gap junctions, the highest concentration in any tissue of the body [101]. Aberrant expression of constituent gap junction proteins, including connexin46 and connexin50, result in cataract and defective lens growth in humans and transgenic mice [141,142,143].

Gap junction-mediated intercellular coupling (GJIC) is higher at the lens equator, relative to either lens pole, and this asymmetry is critical for maintaining lens transparency [144]. Immunofluorescent labeling, and electron microscopy have revealed no quantitative differences in the number of connexins between equatorial and polar fiber cells [145]. Instead, the enhanced GJIC observed at the equator appears to be attributed, in part, to a greater flux through gap junctions within this region [134,146]. Using DCDMLs, FGF-1 or -2 was found to reversibly upregulate GJIC without detectably increasing connexin synthesis or assembly, in an ERK-dependent manner [147]. The ability of FGF to upregulate GJIC is blocked by co-treatment with noggin or highly selective anti-BMP-2, -4 and -7 antibodies [100]. This effect was attributable to inhibition of endogenous lens-derived BMP-4 and -7, that enables FGF-induced ERK-dependent upregulation of GJIC. Although FGF may be necessary for this process, it is not adequate. Inhibition of BMP activity using noggin or chordin abolished the ability of both vitreous humor and FGF to induce GJIC. Furthermore, a selective anti-BMP-7 monoclonal antibody (1B12) inhibited both Smad1 activation and GJIC induced by BMP-7, but not by BMP-2 or BMP-4. This antibody partially blocked the ability of vitreous to upregulate GJIC, and when combined with the anti-BMP-2, 4 antibody, reduced GJIC to control levels. Taken together, these findings again support the importance of the synergistic role of BMP and FGF signal transduction cascades in regulating gap junctional intercellular coupling, an essential postnatal process in lens.

BMP-2, -4 and -7 were shown to increase GJIC in DCDMLs to a comparable extent to that obtained with FGF-treatment. The source of BMP required for increased GJIC was found to originate from the lens and not the vitreous [100], with relatively high concentrations of exogenous BMP-2, -4 and -7 able to promote GJIC in lens cells independent of FGF- or ERK-signaling. At lower, intermediate concentrations, BMPs can stimulate ERK-independent GJIC, but only in the presence of FGF. It is interesting that high levels of BMP-signaling can compensate for the absence of FGF here, but not vice versa. The non-reciprocal crosstalk between FGF- and BMP-signaling pathways is believed to maintain the high levels of GJIC at the lens equator. The high expression of BMP receptors and pSmad1 in the equatorial regions, and declining BMP-signaling in older fiber cells at lens poles, may contribute to the observed reduction in GJIC at these poles, despite the exposure to endogenous FGF [92,93]. Future studies should be aimed at developing in vivo models to better elucidate the role of lens-derived BMPs in regulating GJIC.

## 4. Genetic Mutations in BMPs

Human genetic studies have identified deletions/mutations in four BMP genes, including *bmp-4*, *bmp-7*, *gdf6* (*bmp-13*) and *gdf3*, that are associated with a spectrum of ocular developmental anomalies as well as non-ocular defects [148]. Frameshift and missense mutations in *BMP-4* are found in families with ocular defects, including microphthalmia (small eye), coloboma (incomplete optic fissure closure), myopia, retinal dystrophy and in some cases, anophthalmia (absent eye) [149,150]. Systemic defects varied widely, and typically included structural brain anomalies, macrocephaly, cognitive impairment, diaphragmatic hernia, dental anomalies, polydactyly and short stature [149,150]. Expression studies in human embryos found *BMP-4* in the early stages of eye, brain and digit development, consistent with *BMP-4* mutation phenotypes observed in impacted patients [149]. Moreover, *BMP-4* was localized to the optic vesicle in human embryos, and later restricted to the lens, highlighting its importance in lens/eye development, consistent with earlier reported animal studies [83].

Wyatt et al. (2010) found three heterozygous *BMP-7* mutations, including frameshift, missense and Kozak sequence mutations associated with a spectrum of ocular and non-ocular abnormalities, including anophthalmia, coloboma, cleft palate, developmental delay and skeletal defects [151]. Similarly, mice lacking BMP-7 had severe eye defects including anophthalmia, in addition to kidney and skeletal defects [152]. Incomplete penetrance and variable expressivity were demonstrated in all families, consistent with the variable penetrance of eye abnormalities observed in BMP-7 knockout mice [84,152]. Developmental expression of *BMP-7* in human embryos revealed strong labeling throughout the optic stalk, optic cup and lens vesicle at Carnegie stage (CS)13 and in the retina and lens at CS16, 17 and 19, correlating with the patterns of expression reported in mice [120]. In particular, *bmp-7* expression was elevated at the presumptive fusional edges of the optic fissure, suggestive of a role in fissure closure, and consistent with the presence of coloboma in individuals with BMP-7 mutations.

Numerous studies have reported genetic mutations in *gdf6* in individuals with anophthalmia, coloboma and extraocular anomalies including cleft palate, absent ossicles, polydactyly and skeletal defects, including Klippel-Feil syndrome, hemivertebrae as well as rib and vertebral fusion [153,154,155,156]. Heterozygous missense mutations in *gdf3* also exhibited ocular (microphthalmia and/or coloboma) and skeletal (scoliosis, vertebral fusion, rudimentary 12th rib) defects [157]. Morpholino inhibition of *gdf6a* in zebrafish accurately recapitulated human phenotypes, with ocular defects such as microphthalmia, coloboma, retinal disorganization and hypoplastic optic nerve. Increasing the morpholino impact/dosage resulted in more severe defects of anophthalmia, highlighting the critical role of GDF6 in ocular development [154]. These results were further explored in Xenopus with morpholino inhibition of *gdf6a* resulting in defective lens fiber differentiation, with significant downregulation of lens intrinsic membrane protein 2.3 (*lim2.3*) and crystallin ba2a (*cryba2a*) [87]. These findings indicate that GDF6a may play an important role in later stages of lens development involving terminal differentiation of fiber cells.

Further analyses of larger cohorts manifesting developmental ocular and associated systemic anomalies is important in establishing the full spectrum of defects associated with genetic mutations in BMPs. In turn, this will inform experimental models of transgenic mice and CRISPR knockout studies to elucidate the molecular and genetic basis of normal ocular development and human developmental eye disease. Promising results are emerging with the use of CRISPR technology in the field of bone regeneration. Freitas et al. (2021) used CRISPR-Cas9 to overexpress BMP-9 in mesenchymal stem cells (MSCs) and when these genetically edited cells were injected into rat calvarial bone defects, the BMP-9-overexpressing MSCs were able to repair these defects, with increased bone formation and bone mineral density [158]. Hutchinson et al. (2019) described an innovative methodology using CRISPR/Cas9 to generate endogenous transcriptional reporter cells for the BMP pathway, and this technique could be applied to ocular lens cells to enable future investigations of BMP transcriptional activity in lens development and pathology [159].

## 5. BMPs in Lens Regeneration

Regeneration of the vertebrate lens is a remarkable phenomenon restricted to frogs, salamanders and newts [160,161,162]. Lens regeneration in the adult newt was first observed by Colucci (1891) [163] and independently by Wolff (1895) [164] who provided a more thorough analysis of the process, and hence, this phenomenon has since been referred to as “Wolffian” lens regeneration [165]. Upon removal of the original lens (lentectomy), the process of Wolffian lens regeneration commences with the dedifferentiation of the dorsal iris pigmented epithelium (IPE) [165]. Cells within the IPE become depigmented, expel their melanosomes and these normally mitotically quiescent cells proliferate and transdifferentiate, forming a lens vesicle by day 10 post-lentectomy. The newly formed lens vesicle further differentiates into primary lens fiber cells at 12–16 days. Primary lens fiber cells continue to proliferate from the inner layer while cells from the outer layer of the vesicle differentiate into secondary fibers, and by 25 days, a complete lens is regenerated [166].

Members of the FGF-, BMP- and Wnt-signaling pathways have been implicated in the control of Wolffian lens regeneration [167]. In particular, the dorsal-ventral differences in lens regenerative potency have been partly attributed to spatial differences in BMP-signaling between the dorsal and ventral iris [102]. Grogg et al. (2005) treated newt iris explants (dorsal or ventral) with chordin, or a competitor for the receptor BMPR-IA, to block BMP-signaling, and then re-implanted the iris explants into a host newt. Notably, inhibiting BMP-signaling resulted in the induction of a lens from the normally incompetent ventral iris, with the gene expression profile of the treated ventral irises capable of lens regeneration, similar to that of the dorsal iris during regeneration [102]. This indicated that ventral irises can become “dorsalized” if exposed to the patterns of regulatory events seen in the dorsal iris, conferring the ability to transdifferentiate into lens [102]. Likewise, BMP-7 treatment of dorsal iris explants, and to a lesser extent BMP-4, suppressed its ability to transdifferentiate into lens [102]. This concurs with the established function of BMPs in maintaining ventral identity during embryogenesis, and the resultant dorsalization observed with inhibition of BMP [168].

A different mode of lens regeneration occurs in frogs, in particular in the genus *Xenopus*, specifically *X. laevis*, *X. tropicalis* and *X. borealis* [103,165]. Lens regeneration in Xenopus arises from ectodermal central corneal epithelial cells through a process known as corneal-lens transdifferentiation (CLT) [167]. While newts undergo lens regeneration into adult years, lens regeneration in Xenopus is restricted to larval stages, with a gradual decline in regeneration potential with aging of the tadpole [167]. Freeman described five distinct phases of CLT based on histological analyses in *X. laevis* [169]. At stage 1 (1–2 days post-lentectomy) cells of the inner corneal epithelium undergo a change in morphology from squamous to cuboidal. At stage 2, the cells begin to thicken into the lens placode. At stage 3 (3 days post-lentectomy), a cell aggregate begins to detach from the corneal epithelium and enters the vitreous body. At stage 4, a definitive lens vesicle forms five days post-lentectomy, containing elongated primary lens fiber cells. Finally, a complete lens is observed ten days post-lentectomy, and the cornea reverts to its original squamous epithelial cell phenotype.

The initiation of the CLT process is triggered by exposure of the cornea to factors in the vitreous humor released from the neural retina [170,171]. These factors are normally prevented from reaching the cornea as the lens and corneal endothelium act as simple barriers to the diffusion of these retinal factors [161]. The BMP-, FGF- and Wnt-growth factor signaling pathways have been identified as candidates for induction of lens regeneration in *Xenopus* [167]. Surprisingly, inhibition of BMP-signaling in *Xenopus* induced the opposite effect on lens regeneration compared to the newt [104]. Using a transgenic line of *Xenopus* tadpoles, sustained overexpression of noggin for the first 48 h following lentectomy significantly reduced regeneration [104]. Noggin overexpression appeared to have no effect on the first stage of lens regeneration, as the corneal epithelial cells continued to thicken and transform from a squamous to a cuboidal morphology, similar to wildtype regenerating animals [104]. Prolonged inhibition of BMP-signaling, however, prevented the subsequent progression of cells to transdifferentiate into a new lens and instead, these cells reverted back to a squamous state [104]. The thickened corneal cells were observed to become hypertrophic and died, supporting the role of BMP as a survival factor for cells during regeneration [104]. Using microarrays to identify genes upregulated during the CLT process, the authors identified an increased expression of *Nipsnap1*, a known direct target of BMP. This suggested that *Nipsnap1* was a downstream effector of BMP-signaling, and may facilitate the specification of lens cell fate [104]. The differences in mechanism of BMP-signaling between Wolffian lens regeneration in newts, and CLT in *Xenopus,* may reflect the inherent differences between these distinct regenerative phenomena; however, the extent to which these regenerative processes share specific conserved underlying molecular mechanisms remains unclear.

## 6. BMPs in Cataract Prevention

Although TGFβ and BMPs are members of the same superfamily, they exhibit opposing effects in the lens. TGFβ induces LECs to undergo epithelial-mesenchymal transition (EMT)—a process whereby polarized, immotile LECs acquire apolar, migratory myofibroblastic features bearing morphological and biochemical resemblance to forms of fibrotic cataract, including anterior subcapsular cataract (ASC) and posterior capsular opacification (PCO) [172]. ASC is evident clinically as a dense, white opacity directly beneath the anterior lens capsule [173], while PCO (also known as ‘secondary cataract’ or ‘after-cataract’) manifests as excessive proliferation and migration and EMT of residual LECs over the posterior capsule following cataract surgery [172].

During EMT, LECs abandon their cobblestone morphology and transdifferentiate into characteristic spindle-shaped mesenchymal cells [174,175,176]. In undertaking this phenotypic and morphologic transformation, LECs first experience a loss of tight junction complexes including ZO-1, followed by the loss of E-cadherin, resulting in the redistribution, stabilization and nuclear accumulation of β-catenin [177,178]. LECs undergo a dramatic remodeling of their cytoskeleton, with the de novo expression of α-smooth muscle actin (α-SMA) that is incorporated into the newly formed actin stress fibers [179].

Numerous studies in vitro and in vivo, examining the effects of TGFβ on LECs, have supported the role of this molecule in promoting fibrogenesis via an aberrant transdifferentiation pathway [177,180]. Treatment of LECs with TGFβ results in myofibroblastic transdifferentiation and production of aberrant ECM in rat [181], mouse [182], and human [183] lens epithelial explant systems. In vivo models, including intravitreal injection of active TGFβ into rodent eyes [184], ectopic overexpression of mature TGFβ1 in transgenic mouse lenses [185] and adenoviral gene delivery of TGFβ into mouse anterior chambers [186] have all induced ASC with characteristic EMT features.

There is much evidence that BMP-7 can directly counteract TGFβ-induced EMT in different organ systems [187]. For example, TGFβ promotion of EMT in the kidney, resulting in renal fibrosis, is associated with down-regulation of BMP-7 expression [188]. This EMT response can be reversed with exogenous administration of BMP-7, with restoration of the epithelial phenotype (expression of E-cadherin, ZO-1), and reduction in mesenchymal markers (α-SMA, collagen I, fibronectin and connective tissue growth factor) [189,190]. The activation of Smad1/5 by BMP-7 is reported to block the activation of both Smad3-dependent and Smad-independent pathways, including p38, ERK and MAPKs [188,191,192]. Furthermore, in models of both pulmonary [193] and hepatic [194] fibrosis, adenoviral overexpression of BMP-7 attenuated TGFβ-induced fibrogenic activity via upregulation of inhibitor of differentiation-2 *(Id2)*, a downstream target gene of BMP-7; however, the therapeutic effect of BMP-7 in pulmonary fibrosis is contentious as other studies refute BMP-7’s capacity to reverse or inhibit EMT, suggesting organ specificity for its protective effects [195,196]. Currently, BMP-7, known commercially as osteogenic protein-1 (OP-1) has FDA approval for use in bone repair [197]. Although current animal studies show promising data in the safety and efficacy of systemic administration of BMP-7 for combating fibrosis, it has yet to be applied in human clinical trials.

In the lens, the protective role of BMP-7 has been explored using in vitro and in vivo models (Figure 4). Co-treatment of TGFβ1 and BMP-7 in an α-TN4 murine lens epithelial cell line completely blocked the EMT response, with maintenance of ZO-1 levels and a reduction in α-SMA expression [106]. The inhibitory effect of BMP-7 was diminished with Id2 and Id3 knockdown, highlighting the importance of Id2/3 as nuclear effectors modulating the antagonism between TGFβ and BMP pathways [106]. Work in our laboratory corroborated these findings using a primary rat lens epithelial explant model [108]. We showed that exogenous administration of BMP-7 suppressed TGFβ2-induced EMT by concurrent upregulation of pSmad1/5 and downregulation of pSmad2/3. In addition to the differential Smad upregulation, it is important to note that both BMP-7- and TGFβ-signaling share the common Smad (Smad4) to initiate transcriptional activity and thus, it is possible that their respective antagonistic activity may be attributed to their competition for Smad4. Treatment with TGFβ2 alone suppressed Id2/3 gene expression and addition of BMP-7 restored Id2/3 expression to basal levels indicating a key role for the Id2/3 genes in regulating the inhibitory activity of BMP-7 on TGFβ2-induced lens EMT.

Studies in situ by Saika et al. (2006) investigated the effect of adenoviral-mediated expression of BMP-7, Id2 or Id3 in a mouse lens capsular injury-induced model of EMT [107]. Lens capsular injury induced low expression levels of endogenous BMP-7 mRNA and protein, that subsequently upregulated expression of Id2 and Id3 [107]. Gene transfer of BMP-7, Id2 or Id3 effectively delayed injury-induced EMT by maintenance of the epithelial phenotype and reductions in EMT markers (α-SMA and collagen type VI) [107]. This suppression of EMT was accompanied by a reduction in Smad2 phosphorylation and upregulation of pSmad1/5/8. Although this gene transfer attenuated the EMT response, its inhibitory effect did not last beyond 10 days, with elongated fibroblastic cells present despite the BMP-7, Id2 and Id3 expression persisting. Although BMP-7 has been shown to effectively antagonize TGFβ using in vitro lens epithelial cell models, it merely delays the progress of EMT in lens in vivo. It is likely that the combined activity of BMP-7 and various inherent growth factors in the aqueous humor, may impact its efficacy. Continued research is required to elucidate the conditions responsible for enhancing or diminishing the inhibitory capabilities of BMP-7. Work in bone formation highlighted a role for Ski and SnoN, transcriptional co-factors, in regulating the antagonistic relationship between TGFβ- and BMP-signaling [198]. Specifically, the authors showed that TGFβ1 blocked both BMP-2 and BMP-7 Smad-signaling in primary human osteoblasts by upregulating Ski and SnoN and increasing histone deacetylase (HDAC) activity. Thus, adding a HDAC inhibitor such as valproic acid as an adjunct to BMP therapy, may improve the efficacy of BMP therapy to further suppress TGFβ activity.

More recently, BMP-4 has also emerged as a potential inhibitor of lens EMT. Work in our laboratory showed that BMP-4 can block TGFβ2-induced EMT in rat lens epithelial explants by suppressing Smad2/3 nuclear translocation [109]. The protective effect of BMP-4 has been further demonstrated in the human lens epithelial cell lines (HLE-B3), where exogenous addition of BMP-4 blocked apoptosis of lens epithelial cells under H_2_O_2_-induced oxidative stress [110]. Intriguingly, small molecule agonists of BMPs, ventromorphins, were unable to suppress TGFβ2-induced lens EMT in rat lens explants, highlighting that not all approaches to promote BMP-signaling can block TGFβ2-induced lens EMT [109]. Rather, particular conditions may exist that favor the efficacy of certain BMP isoforms in blocking TGFβ2 activity. Further unravelling of these intricate and nuanced differences will enable us to develop more effective, targeted novel therapies to combat fibrotic cataract.

## 7. Conclusions and Future Directions

Although important advances have been made in elucidating the role of BMPs and BMP-signaling in the lens, it is clear from this review that there are still significant gaps in our understanding. Specifically, detailed investigations of spatiotemporal expression patterns of BMPs and their receptors in embryonic lens development also need to be further explored in adult lens. Moreover, the majority of studies on BMPs have utilized animal models, with very few human studies reported, with no current clinical trials for BMPs, highlighting the important research direction for translating animal research to human therapeutics.

Significant progress has been made in characterizing the canonical and non-canonical BMP-signaling pathways in non-ocular tissues; however, many of these advances are yet to be explored in the lens. Do specific BMP isoforms or receptors play more prominent roles in certain aspects of lens development, regeneration or cataract prevention? If so, what are the precise intracellular and extracellular regulators that activate certain lens programs, and suppress alternate programs? Are there additional regulatory mechanisms, such as post-translational modifications or epigenetic changes, that dictate the cellular response to BMPs in the lens? Are there regulatory signals upstream of BMP-signaling and how do they ultimately converge to exert the numerous biological roles of BMPs?

Since the BMP family consists of multiple ligands and receptors that interact promiscuously with each other, a multitude of distinct signaling complexes can be generated [199]. Using mathematical modeling and computational analysis, Antebi et al. (2017) showed that cells are able to perform complex computations with their given set of ligands and receptors, based on the specific functions of ligand combinations, rather than base their activity purely on ligand abundance and binding affinities. Clearly, there is much versatility and regulatory flexibility in BMP-signaling. Further characterization of the combinations of ligands, and introducing the effects of diffusible inhibitors, such as noggin and chordin into the computational analysis, will enable a better understanding of lens embryogenesis and development that are already dependent on many different BMP ligands, receptors and modulators, all expressed in spatially and temporally overlapping patterns. Continued research in the role of BMPs in the lens will not only help elucidate key developmental lens processes but also open avenues for the development of novel therapeutic strategies for lens regeneration and possibly cataract prevention.

## Figures and Tables

**Figure 1 cells-10-02604-f001:**
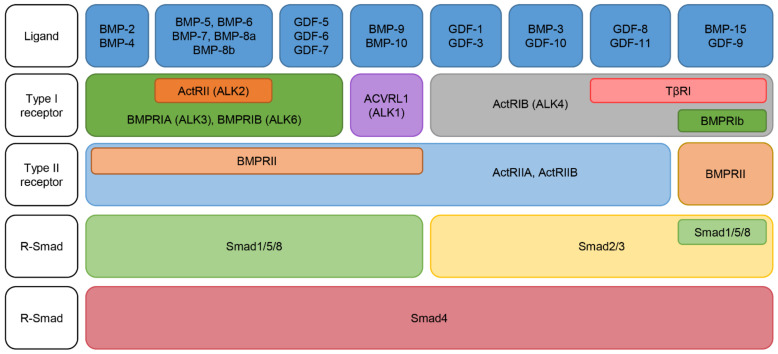
Bone morphogenetic protein (BMP) ligands and receptors. Different BMP ligands bind to different type I and II BMP receptors to activate the canonical Smad-signaling pathway involving the receptor regulated-Smads (R-Smads) and the common Smad (Co-Smad). GDF (growth differentiation factor); ALK (activin-like kinase); ActR (activin receptor).

**Figure 2 cells-10-02604-f002:**
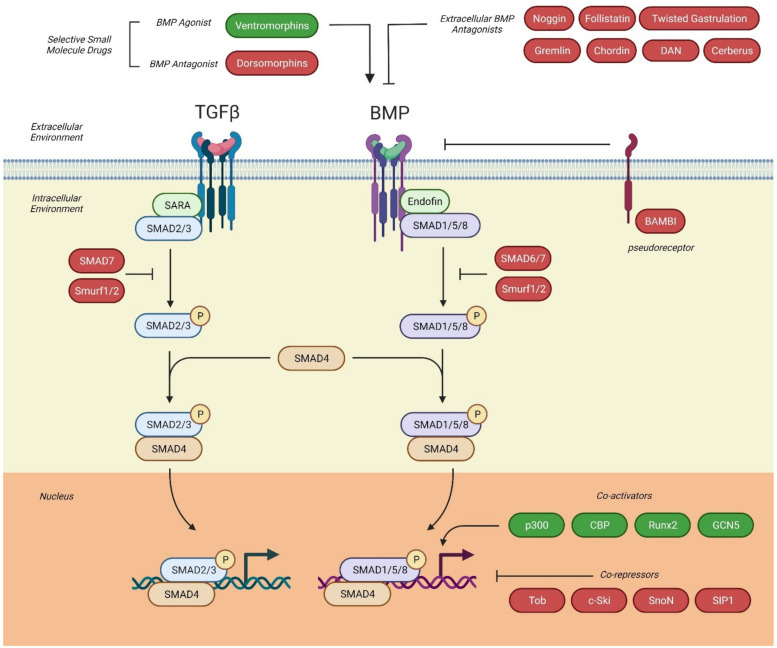
Transforming growth factor beta (TGFβ) and bone morphogenetic protein (BMP) receptor signal transduction. TGFβ and BMP bind to their respective type I and II receptors to activate the downstream canonical Smad-signaling to initiate gene transcription by binding various co-activators and co-repressors. While TGFβ activates Smad2/3 and BMP activates Smad1/5/8, both require the common Smad, Smad4, to form a complex for nuclear translocation. Inhibitory Smads (Smad6/7) and Smurf1/2 act as intracellular negative regulators of the TGFβ- and/or BMP-pathway. Several extracellular BMP antagonists/agonists and the pseudo-receptor, BAMBI, regulate BMP-signaling.

**Figure 3 cells-10-02604-f003:**
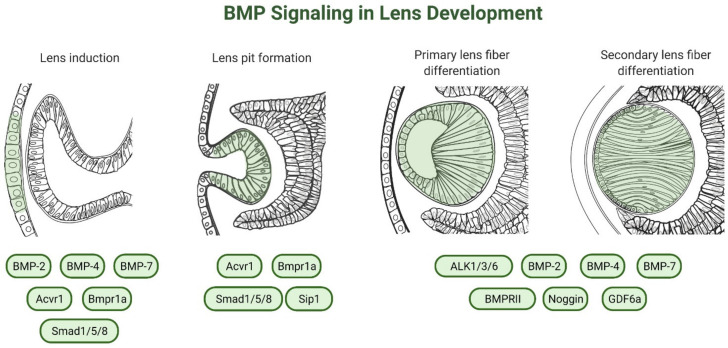
Involvement of bone morphogenetic protein (BMP) signaling in lens development.

**Figure 4 cells-10-02604-f004:**
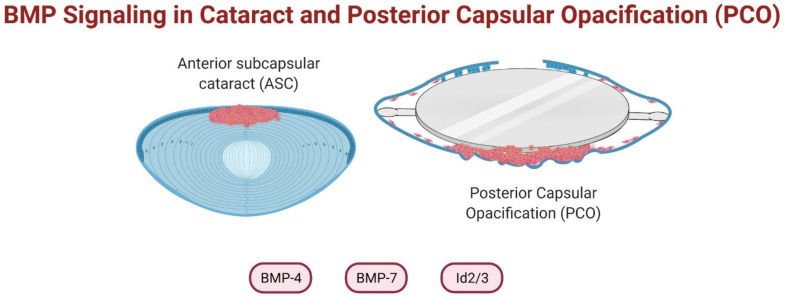
Involvement of bone morphogenetic protein (BMP) antagonistic signaling in anterior subcapsular cataract (ASC) and posterior capsular opacification (PCO) progression.

**Table 1 cells-10-02604-t001:** Summary of the studies investigating the role of BMPs in lens induction, lens fiber differentiation, gap junction-mediated communication, lens regeneration and cataract prevention in various experimental models.

Author (Year)	Experimental Model	BMPs Investigated
**Lens Induction**		
Luo et al. (1995) [27]	In vivo mouse	BMP-7
Furuta et al. (1998) [83]	In vivo mouse	BMP-4
Wawersik et al. (1999) [84]	In vivo mouse	BMP-7
Zhao et al. (2002) [85]	In vivo mouse	BMP-7, noggin
Sjödal et al. (2007) [86]	In vivo chick	BMP-4
French et al. (2009) [87]	In vivo zebrafish	BMP-4, GDF6a
Rajagopal et al. (2009) [88]	In vivo mouse	BMP receptor Acvr1 and Bmpr1a
Huang et al. (2015) [89]	In vivo chick, in vivo mouse	BMP-7, Acvr1, Bmpr1a
**Lens Fiber Differentiation**		
Hung et al. (2002) [90]	In vivo mouse	BMP-7
Faber et al. (2002) [91]	In vivo mouse	Bmpr1b
Belecky-Adams et al. (2002) [92]	In vivo chick	BMP-2, BMP-4, BMP-7, noggin
de Iongh et al. (2004) [93]	In vivo mouse, rat lenses	ActRIIA, ActRIIB, BmprII, ALK3
Jarrin et al. (2004) [94]	In vivo chick	Noggin
Pan et al. (2006) [95]	In vivo mouse	BMP-4
Boswell et al. (2008) [81]	In vitro embryonic chick	BMP-2, BMP-4, BMP-7, noggin
Rajagopal et al. (2009) [88]	In vivo mouse	BMP receptor Acvr1
Pandit et al. (2011) [96]	In vitro in vivo chick	BMP-4
Wiley et al. (2011) [97]	In vivo mouse	BMP receptor Acvr1
Jidigam et al. (2015) [98]	In vivo chick	BMP-4, BMP-7
Boswell et al. (2015) [99]	In vitro embryonic chick	BMP-2, BMP-4, BMP-7, noggin
**Gap-junction Mediated Communication**		
Boswell et al. (2008) [100]	In vitro embryonic chick	BMP-2, BMP-4, BMP-7, noggin
Boswell et al. (2009) [101]	In vitro embryonic chick	BMP-4
**Lens Regeneration**		
Grogg et al. (2005) [102]	In vivo newt	BMP-4, BMP-7, chordin, Bmpr1a
Kurata et al. (2001) [103]	Xenopus	BMP-4
Day and Beck (2011) [104]	Xenopus	Noggin, *Nipsnap1*
Yang et al. (2010) [105]	Human embryonic stem cells	BMP-4, BMP-7, noggin
**Cataract Prevention**		
Kowanetz et al. (2004) [106]	Mouse epithelial cell line	BMP-7, Id2, Id3
Saika et al. (2006) [107]	In vivo mouse	BMP-7, Id2, Id3
Shu et al. (2017) [108]	In vitro rat lens explant	BMP-7, Id2, Id3
Shu et al. (2021) [109]	In vitro rat lens explant	BMP-4, ventromorphins
Du et al. (2021) [110]	HLE-B3 human lens cell line	BMP-4

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
