# Peer review of "Insights into Bone Morphogenetic Protein—(BMP-) Signaling in Ocular Lens Biology and Pathology"

_cells, 2021, doi:10.3390/cells10102604_

Round 1

Reviewer 1 Report

The manuscript by Shu and Lovicu „Insights into Bone Morphogenetic Protein- (BMP-) Signaling in Ocular Lens Biology and Pathology“ is a well written and constructed review on BMPs with particular reference to lens biology and pathology. The figures supporting the statements covering BMP ligands and receptors, as well as signal transduction molecules are well designed.

The major criticism refers to the BMP historical overview which is not appropriately referenced using, instead of original discoveries, reviews written by scientists who did not participate in the discovery of BMP members, signaling and biology, as substitutes for major discoveries in the field. This is inappropriate referencing which should be corrected to improve this otherwise well elaborated BMP review.

The suggestions for corrections are indicated along with line numbers at the right side of the text:

  • Line 30: Reference 2 should be replaced or supplemented with Wozney et al, Science 1988. Since these references cover only BMP1, BMP2, BMP3 and BMP4 (BMP1 is not a real BMP and is elaborated in the text later), other BMPs discovered in parallel should be mentioned here, covered by Ozkaynak et al. EMBO J 1990 and JBC 1992.
  • Line 33: Reference 3 used as a review for elaborating initial discovery and pleiotropic effect of BMPs is not an appropriate source since it contains inappropriate referencing to original discoveries and missing important findings in the BMP field.
  • Line 35: Reference 4 is not appropriate for presenting BMP members and their role in embryogenesis and homeostasis since it refers to their periodontal use.
  • Line 55: References 3 and 7 are inappropriate for the statement about BMP structure/function relationship.
  • Line 61: Reference 9 is inappropriate for the statement about BMP cleavage and separation of the mature protein portion from its prodomain, originally presented by Jones WK et al. Growth Factors 1994 referring to BMP7/OP-1
  • Line 74: BMP12/13/14 (GDF5/6/7) should be in opposite order since GDF5=BMP14. Here it should be references that the subgroup BMP14/13/12 is identical to GDF 5/6/7 as well as named CDMP1/2/3 (cartilage derived morphogenetic proteins) published in parallel by groups of Frank Luyten, Hari Reddi and Malcolm Moose in JBC 1994 (first author Chang S et al). Several other important members of the BMP GDF are missing, including GDF15, originally named PDF (Paralkar et al. JBC 1998).
  • Line 77: Reference 13 by Karsenty has nothing to do with discovery of BMP1 as a procollagen C proteinase and should be replaced by Kessler et al. Science 1996.
  • Line 86: The BMP7 expression in the kidney and several other organs has been discovered by Helder et al. J Histochem Cytochem 1995 and Bosukonda et al. Kidney Int 2000. which should be appropriately acknowledged.
  • Line 93: The role of BMP6 in iron regulation has been discovered by Andriopoulos et al. Nat Genet 2009 and should be added to ref. 25.
  • Line 102: ref 11 Here the work of Peter ten Dijke and CH Heldin should also be cited as they were the pioneers in cloning the type I receptors (ten Dijke et al. J Biol Chem 1994; Souchelnytskyi S et al. EMBO J 1996).
  • Line 113, ref. 29: Here the correct primary paper is Nohe A et al., JBC 2002.
  • Line 117/118: The outcome of binding to either the individual or to the complex of both types of receptor is different, described in both Nohe et al and independently with other experimental set ups in Guzman et al (Nohe et al, JBC 2002; Guzman et al JBC 2012).
  • Lines 126-129: ref 32 The reference Hartung et al., MCB 2006 and Nohe et al, JBC 2002 should be added.
  • Lines 197-180: The cross-talk to the PI3K pathway is missing
  • Line 187: ref 41 This is not the correct reference for this statement: the relevant work by J. Massague and the work by E. de Robertis should be included.
  • Line 200: The term “isoforms” is not correct here; the authors rather mean members of the family but not isoforms since they are encoded by different genes.
  • Line 204: ref 48 There are more examples to this published; references missing.
  • Line 205: Ref 49 BMP6 and BMP9 insensitivity has been originally discovered by Song K. et al. JBC 2010 285:12169-80.; ref. 49 was published later but is important and could remain as a reference.
  • Line 208: ref. 44 Cordin/BMP2/4 complex cleaved by BMP1 has been originally described by Ge G and Greenspan DS. J Cell Biol 2006.; ref. 44 is inappropriate – a review from a dermatology journal.
  • Line 212-215: There are more coreceptors to be mentioned here, e.g. RGMs.
  • Line 245 - Table 1: Gives an overview of BMPs in lens, however, the references are not covered by numbers used in the reference list and therefore it is impossible to follow them in the text.

Reviewer 2 Report

This is a clearly written and logically presented review paper on the roles of the BMP signaling family in lens development, regeneration and cataract.  It has sourced material from a wide array of studies and appropriately described data from a range of species to provide very clear insights as well as identified areas for further investigation.

It was a pleasure to read and I have only minor editorial issues to raise:

Figure 1:  One of the Type II receptor is labelled as ActRIA, presumably this should be ACTRIIA.  Please correct all protein names to all capitals.  Even nomenclature of murine proteins is now all capitals. Since human and mice are the predominant species discussed this should be the most appropriate nomenclature for proteins in this figure.

In various places the authors forget to insert appropriate citations. While they mention in an introductory sentence, they often forget to include appropriate citations in subsequent descriptions. I have identified some instances below by line number, but the authors should check the manuscript in greater detail to ensure there are no other sections where citations are lacking.

  • Lines 299-302: attribute the appropriate papers for the chick and mouse work.
  • Lines 299-302: which paper demonstrated this difference between Bmp7 and Bmp4 expression?
  • Lines 527-536: probably needs re-iteration of the Beebe citation (#97) at some point to clarify that all these experiments were conducted by the Beebe group and not just the change in expression of TP53?

In line 351, “is fitting with” seems cumbersome; suggest “is consistent with”.

In line 368, “To note” seems cumbersome; suggest “Notably”.

Reviewer 3 Report

The review entitled ' Insights into Bone Morphogenetic Protein- (BMP-) Signaling in Ocular Lens Biology and Pathology’ describes the multifaceted role of BMP signalling, the molecular mechanisms that are involved in the Ocular biology and its role in lens regeneration.

The review article is very interesting and describes in detail the mechanisms that are involved. However, the authors could improve the article to broaden the scope of readership.

  1. The authors nicely describe about the lens development in Section 3. A figure summarising the multistep process of various BMPs involved in lens development is crucial to the review (in relation to section 3).
  2. In Section 4, the authors describe about the genetic mutations that are involved in ocular developmental anomalies, can the authors discuss the potential therapeutic interventions that are possible. In this context the possibility to use CRISPR technology to generate knockouts and if any published research that are available currently and also the latest developments in the field.
  3. The last section related to the effect of BMP7 in lens cataract prevention is interesting. Is there a competitive signalling between TGFbeta and BMP signalling in this context? Can the authors comment about it?
  4. They should highlight if any clinical trials are currently undergoing ?
  5. They should include an overview figure about the therapeutic interventions that are available and emerging targets that needs further exploration.
